# Temporal Change in Anterior Chamber Depth after Combined Vitrectomy and Cataract Surgery Using Different Sizes of Intraocular Lens

**DOI:** 10.3390/jcm11216430

**Published:** 2022-10-30

**Authors:** Yoshiaki Takahashi, Takao Hirano, Marie Nakamura, Yoshiaki Chiku, Ken Hoshiyama, Satoko Akahane, Keita Akahane, Toshinori Murata

**Affiliations:** 1Department of Ophthalmology, Shinshu University School of Medicine, Matsumoto 390-8621, Japan; 2Nagano Red Cross Hospital, Nagano 380-0928, Japan; 3Suwa Red Cross Hospital, Suwa 392-8510, Japan

**Keywords:** anterior chamber depth, phacovitrectomy, anterior segment optical coherence tomography

## Abstract

This study aimed to investigate the temporal changes in the anterior chamber depth (ACD) and refractive prediction error (RPE) of 6 and 7 mm diameter intraocular lenses (IOLs) after cataract surgery or phacovitrectomy with or without sulfur hexafluoride (SF6) gas. We reviewed 120 eyes (120 patients) and divided them into six groups: three groups of cataract surgery alone and phacovitrectomy (with or without SF6), which were further divided according to IOL diameter (6 and 7 mm) used. We examined the ACD and IOL position using a swept-source anterior segment optical coherence tomograph at 1 day, 1 week, and 1 month postoperatively; and the RPE at 1 month postoperatively. The ACD and IOL position at postoperative day 1 in the SF6 injection groups were significantly smaller than those in the other groups (*p* < 0.01). At 1 week, the ACD and IOL position of the 6 mm IOL SF6 injection group was smaller than those in the other groups (*p* < 0.01); thus, complications such as synechia or pupillary capture should be considered in the early postopera-tive period of phacovitrectomy with SF6, especially in the 6 mm IOL. The ACD, IOL position, and RPE at 1 month did not significantly differ among the groups, regardless of the IOL diameter or SF6. In the SF6 injection groups, the ACD and IOL position gradually deepened with less gas.

## 1. Introduction

Pars plana vitrectomy combined with phacoemulsification, aspiration, and intraocular lens (IOL) implantation (phacovitrectomy) can prevent secondary cataracts that may occur with vitrectomy alone in a phakic eye; thus, reducing patient burden and cost [1,2]. In phacovitrectomy, as in cataract surgery, the best corrected visual acuity (BCVA) is needed. To achieve this, the refractive prediction error (RPE) should be low. However, many reports have shown that the RPE after phacovitrectomy is approximately −0.50 diopter (D) [3,4]. This myopic shift is attributed to several factors, such as the difference in refractive index between the vitreous and aqueous humor [5], different types of IOL, and use of gas tamponade [6,7,8,9,10]. To evaluate the influence of the types of IOL and the use of gas tamponade on RPE, anterior segment information should be measured accurately. In recent years, swept-source anterior segment optical coherence tomography (SS-ASOCT) has been clinically applied. CASIA2 (Tomey, Nagoya, Japan), an SS-ASOCT, can acquire three-dimensional anterior segment information, including both corner angles in a range of 16 × 16 mm, in approximately 2 s by taking 128 consecutive B-scan images in the axial rotation direction and is expected to measure ACD with a high accuracy [11]. A study that used CASIA2 reported that myopic refractive error occurred due to anterior fixation of the IOL even after the gas had resolved in eyes undergoing phacovitrectomy with gas tamponade [12]. However, this study did not examine the effect of IOL size. Therefore, differences in IOL behavior and postoperative refraction according to the size of the lens, presence of gas, and differences from cataract surgery alone are expected. This study examined the degree of IOL position shift in cataract surgery alone, and with and without gas after phacovitrectomy; and compared its effect on postoperative refraction with previous reports. We investigated the changes in postoperative ACD and RPE in cataract surgery alone, and in phacovitrectomy with and without gas tamponade using CASIA2, including the differences seen between 6 and 7 mm IOLs.

## 2. Materials and Methods

### 2.1. Study Design and Patients

Patients who underwent phacovitrectomy or cataract surgery alone between April 2020 and July 2022 at Shinshu University Hospital or Suwa Red Cross Hospital were included. We consecutively recruited patients until the prescribed number of cases was reached within the time frame. The exclusion criteria were uncompleted for the operation because of capsule rupture, cases in which the IOL could not be fixed in the capsule, corneal disease such as keratoconus, re-operation required after surgery, and postoperative synechia or dislocation that prevents accurate ACD measurement.

### 2.2. Preoperative Examinations

All the patients underwent comprehensive ophthalmologic examinations before the operation, including refraction, BCVA, intraocular pressure, corneal endothelial cell counts, fundus imaging, and spectral-domain optical coherence tomography. Axial length was measured using the IOL Master (Carl Zeiss Meditec, Oberkochen, Germany). We calculated IOL power using the SRK/T formula and the A constant recommended by the manufacturer. Anterior segment evaluation was conducted using CASIA2.

### 2.3. Anterior Segment Evaluation

CASIA2 has a resolution of 10 µm in the horizontal direction and 30 µm in the vertical direction. CASIA2 can gather the image up to the posterior surface of the lens [13]. We used CASIA2 to measure the ACD, lens thickness, central corneal thickness, and lens tilt. The ACD was defined as the distance from the posterior surface of the cornea to the front of the lens (Figure A1). In the case of an obviously wrong measurement area, we corrected the alignment manually. The IOL position, considered to be less affected by individual factors such as axial length, was calculated by the change in anterior chamber depth divided by lens thickness, as previously reported [12].

### 2.4. Surgical Procedure

In cataract surgery, a 2.4 mm scleral incision is made using a 6 mm IOL, whereas a 3.0 mm scleral incision is made using a 7 mm IOL. In the case of cataract surgery alone, phacoemulsification was performed with Centurion (Alcon Laboratories, Inc., Fort Worth, TX, USA); in the case of phacovitrectomy, a 25-gauge pars plana vitrectomy with constellation (Alcon Laboratories, Inc.) was performed. The IOL was fixed intracapsular in all cases. The type of lenses used were XY-1^®^ (Hoya, Tokyo, Japan), Clareon^®^ (Alcon Laboratories, Inc.), YP2.2R^®^ (Kowa, Tokyo, Japan), Tecnis Eyhans^®^ Optiblue^®^ (Johnson & Johnson Vision, Santa Ana, CA, USA) in 6 mm diameter lenses, and X-70^®^ (Santen, Osaka, Japan) in 7 mm diameter lenses. The IOL features are shown in Table 1. The phacovitrectomy procedure included: (1) core vitrectomy; (2) peripheral vitrectomy with compression of the sclera; (3) removal of the epiretinal membrane (ERM) and internal limiting membrane in necessary cases; and (4) replacement of 20% sulfur hexafluoride (SF6) in the macular hole (MH) or retinal detachment (RD), as necessary. Postoperatively, SF6-replaced patients were placed in the prone position for 1 week to 2 weeks.

### 2.5. Postoperative Examinations

The ACD and IOL position were measured using CASIA2 at 1 day, 1 week, and 1 month postoperatively. The refraction value and BCVA were evaluated at 1 month postoperatively. We defined RPE as the postoperative spherical equivalent minus the SRK/T predicted refraction value.

### 2.6. Analysis

The following six groups were defined: 6 mm diameter IOL with SF6, 6 mm diameter IOL without SF6, 7 mm diameter IOL with SF6, 7 mm diameter IOL without SF6, 6 mm diameter IOL cataract surgery alone, and 7 mm diameter IOL cataract surgery alone. We performed statistical analysis on the mean values of the six groups for ACD at all the evaluations and for the IOL position at each postoperative evaluation. We compared the RPEs at 1 month postoperatively among the six groups. Statistical analyses were performed using Python 3.9.12, scipy.stats module (Phyton, Wilmington, DE, USA). Statistical significance was set at *p* < 0.05. Comparisons among multiple groups were made using the Friedman test when there was correspondence and the Kruskal-Wallis test when there was no correspondence. If the difference was significant, individual means were compared using the Bonferroni correction method. Comparisons between the two groups used the Wilcoxon signed rank test for correspondence and the Mann–Whitney test for non-correspondence. Continuous values are expressed as mean ± standard deviation.

## 3. Results

### 3.1. Demographic Information

There were 20 cases in each of the six groups. We investigated 120 eyes of 120 patients. The mean age of all the patients was 67.2 ± 9.9 years (range, 36–86 years). The mean axial length was 24.2 ± 1.57 mm, the mean lens thickness was 4.56 ± 0.41 mm, and the mean preoperative BCVA (logMAR) was 0.53 ± 0.54. The details of each group are shown in Table 2. The diseases being treated in the 6 mm IOL with SF6 injection group were RD (9 eyes), MH (6 eyes), ERM (3 eyes), and retinoschisis (2 eyes); and those in the 7 mm IOL with SF6 injection group were RD (9 eyes), MH (9 eyes), and proliferative diabetic retinopathy (PDR; 2 eyes). The diseases in the 6 mm IOL without SF6 injection group were ERM (17 eyes), PDR (2 eyes), and vitreous opacity (1 eye); and those in the 7 mm IOL without SF6 injection group were ERM (12 eyes), PDR (7 eyes), and vitreous hemorrhage (1 eye).

### 3.2. Preoperative and Postoperative ACD Changes

The ACDs at 1 day postoperatively were significantly larger in all the groups compared to the preoperative ACDs (*p <* 0.01). The mean ACDs in the 6 mm IOL cataract surgery-alone group were 2.78 ± 0.39 mm preoperatively, 4.27 ± 0.27 mm at 1 day postoperatively, 4.17 ± 0.28 mm at 1 week postoperatively, and 4.15 ± 0.30 mm at 1 month postoperatively, whereas in the 7 mm IOL cataract surgery-alone group, the ACDs were 2.80 ± 0.28, 4.33 ± 0.28, 4.17 ± 0.20, and 4.14 ± 0.21 mm, respectively. In the 6 mm IOL with SF6 group, the ACDs were 2.92 ± 0.40, 3.29 ± 0.49, 3.74 ± 0.36, and 4.16 ± 0.31 mm, respectively. In the 7 mm IOL with SF6 group, the ACDs were 2.85 ± 0.33, 3.51 ± 0.26, 4.06 ± 0.25, and 4.18 ± 0.27 mm, respectively. In the 6 mm IOL without SF6 group, the ACDs were 2.61 ± 0.36, 4.14 ± 0.27, 4.10 ± 0.29, and 4.04 ± 0.26 mm, respectively. In the 7 mm IOL without SF6 group, the ACDs were 2.77 ± 0.24, 4.17 ± 0.28, 4.04 ± 0.17, and 4.02 ± 0.14 mm, respectively.

Although the ACDs gradually decreased in four groups (i.e., 6 and 7 mm IOL cataract surgery-alone groups, and 6 and 7 mm IOL without SF6 groups) postoperatively, the ACDs gradually increased in two groups (i.e., 6 and 7 mm IOL with SF6 groups) postoperatively.

In multiple comparisons, the ACDs of three groups (i.e., 6 mm IOL cataract surgery-alone group, and 6 and 7 mm IOL without SF6 groups) were smaller at 1 week postoperatively than at 1 day postoperatively (*p <* 0.017 after Bonferroni correction); however, there was no difference between 1 week and 1 month postoperatively (*p =* 0.17, 0.07, and 0.13 after Bonferroni correction). In the 7 mm IOL cataract surgery-alone group, the ACD was largest at 1 day postoperatively, significantly smaller at 1 week postoperatively than at 1 day postoperatively (*p <* 0.017 after Bonferroni correction), and significantly smaller at 1 month postoperatively than at 1 week postoperatively (*p <* 0.017 after Bonferroni correction). In both 6 and 7 mm IOL with SF6 groups, the ACDs were smallest at 1 day postoperatively, significantly larger at 1 week postoperatively than at 1 day postoperatively (*p <* 0.017 after Bonferroni correction), and significantly larger at 1 month postoperatively than at 1 week postoperatively (*p <* 0.017 after Bonferroni correction).

The 6 and 7 mm IOL with SF6 groups had a significantly smaller ACD than the other groups at 1 day postoperatively (*p <* 0.01: Kruskal-Wallis test, *p <* 0.003 after Bonferroni correction). The 6 mm IOL with SF6 group had a smaller ACD than the other groups at 1 week postoperatively (*p*
*<* 0.01: Kruskal-Wallis test, *p <* 0.003 after Bonferroni correction), and there was no difference in the ACDs of all the groups at 1 month postoperatively (*p =* 0.26) (Figure 1 and Table 3).

### 3.3. IOL Position Changes

The mean IOL positions in the 6 mm IOL cataract surgery-alone group were 0.33 ± 0.06 at 1 day postoperatively, 0.31 ± 0.05 at 1 week postoperatively, and 0.30 ± 0.05 at 1 month postoperatively. In the 7 mm IOL cataract surgery-alone group, the IOL positions were 0.33 ± 0.03, 0.30 ± 0.03, and 0.29 ± 0.03, respectively. In the 6 mm IOL with SF6 group, the IOL positions were 0.08 ± 0.09, 0.18 ± 0.09, and 0.28 ± 0.02, respectively. In the 7 mm IOL with SF6 group, the IOL positions were 0.14 ± 0.05, 0.27 ± 0.04, and 0.30 ± 0.04. In the 6 mm IOL without SF6 group, the IOL positions were 0.32 ± 0.04, 0.29 ± 0.02, and 0.30 ± 0.02, respectively. In the 7 mm IOL without SF6 group, the IOL positions were 0.32 ± 0.07, 0.29 ± 0.05, and 0.28 ± 0.04, respectively. The changes were similar to those of the ACDs in all the groups.

Although the IOL positions gradually decreased in four groups (i.e., 6 and 7 mm IOL cataract surgery-alone groups, and 6 and 7 mm IOL without SF6 groups) postoperatively, the IOL positions gradually increased in two groups (i.e., 6 and 7 mm IOL with SF6 groups) postoperatively.

In multiple comparisons, the IOL positions of three groups (i.e., 6 mm IOL cataract surgery-alone group, and 6 and 7 mm IOL without SF6 groups) were smaller at 1 week postoperatively than at 1 day postoperatively (*p <* 0.017 after Bonferroni correction); however, there was no difference between 1 week and 1 month postoperatively (*p =* 0.18, 0.11, and 0.14 after Bonferroni correction). In the 7 mm IOL cataract surgery-alone group, the IOL position was significantly smaller at 1 week postoperatively than at 1 day postoperatively (*p <* 0.017 after Bonferroni correction), and significantly smaller at 1 month postoperatively than at 1 week postoperatively (*p <* 0.017 after Bonferroni correction). In the 6 and 7 mm IOL with SF6 groups, the IOL positions were smallest at 1 day postoperatively, significantly larger at 1 week postoperatively than at 1 day postoperatively (*p <* 0.017 after Bonferroni correction), and significantly larger at 1 month postoperatively than at 1 week postoperatively (*p <* 0.017 after Bonferroni correction).

The 6 and 7 mm IOL with SF6 groups had a significantly smaller IOL position than the other groups at 1 day postoperatively (*p <* 0.01: Kruskal-Wallis test, *p <* 0.003 after Bonferroni correction). The 6 mm IOL with SF6 group had a smaller IOL position than the other groups at 1 week postoperatively (*p <* 0.01: Kruskal-Wallis test, *p <* 0.003 after Bonferroni correction), and there was no difference in IOL position among all the groups at 1 month postoperatively (*p =* 0.45) (Figure 2 and Table 4).

### 3.4. BCVA and RPE Outcomes

Overall, the mean BCVAs were 0.53 ± 0.54 preoperatively and 0.21 ± 0.31 postoperatively, with a significant improvement (*p <* 0.01). The preoperative and postoperative mean BCVAs in each group were 0.54 ± 0.59 and 0.15 ± 0.25, respectively, in the 6 mm IOL cataract surgery-alone group; 0.36 ± 0.23 and 0.21 ± 0.30, respectively, in the 7 mm IOL cataract surgery-alone group; 0.41 ± 0.46 and 0.11 ± 0.19, respectively, in the 6 mm IOL with SF6 group; 0.61 ± 0.50 and 0.27 ± 0.41, respectively, in the 7 mm IOL with SF6 group; 0.37 ± 0.23 and 0.27 ± 0.33, respectively, in the 6 mm IOL without SF6 group; and 0.92 ± 0.81 and 0.35 ± 0.29, respectively, in the 7 mm IOL without SF6 group. In the 6 mm IOL without SF6 group, there was no significant difference between the preoperative and postoperative BCVAs (*p =* 0.11). In the other five groups, there was a significant improvement of BCVA (*p <* 0.01).

The mean RPEs were −0.10 ± 0.25 in the 6 mm IOL cataract surgery-alone group, −0.29 ± 0.83 in the 7 mm IOL cataract surgery-alone group, −0.11 ± 0.71 in the 6 mm IOL with SF6 group, −0.35 ± 0.75 in the 7 mm IOL with SF6 group, −0.33 ± 0.56 in the 6 mm IOL without SF6 group, and −0.47 ± 0.94 in the 7 mm IOL without SF6 group. There was no significant difference in each group (*p =* 0.70).

## 4. Discussion

We compared the ACDs and refractive outcomes after cataract surgery or phacovitrectomy. A comparison of the six groups using CASIA2 showed no differences in ACD, IOL position, or RPE in all the groups at 1 month postoperatively. However, the trends of ACD and IOL position up to 1 month after surgery differed significantly depending on the IOL diameter or SF6 use.

In all the observations, the postoperative ACDs were significantly larger than the preoperative ACDs in all the groups due to lens removal. The ACD and IOL position trends were completely similar throughout the whole period. In the 6 and 7 mm IOL cataract surgery-alone groups and the 6 and 7 mm IOL without SF6 injection groups, postoperative ACDs did not differ at any observation period among the groups. However, these groups tended to have a decrease in ACD from 1 day to 1 week postoperatively. This shows that it took 1 week to 1 month for ACD to stabilize. In the 7 mm IOL cataract surgery-alone group, the ACD and IOL position were significantly smaller at 1 month than at 1 week postoperatively. In the 6 mm IOL cataract surgery-alone group, and the 6 and 7 mm IOL without SF6 groups, there was no significant difference between 1 week and 1 month postoperatively; however, the ACD was slightly smaller. A report showed that it takes 2 weeks for the ACD to stabilize after cataract surgery [14]; therefore, we considered that fluctuations in the ACD occur at least 2 weeks to 1 month postoperatively. Regardless of the postoperative duration, there was no significant difference in the ACD in these four groups. The lack of difference between the cataract surgery-alone group and the without SF6 group was consistent with previous reports [15]. We examined the IOL positions that were not affected by axial length, and there were no differences in trend between the ACD and the IOL positions. We considered that the similar trend of the ACD and IOL positions is due to the lack of difference in axial length among the six groups in this study.

In the SF6 groups, the ACD and IOL position showed a tendency to increase over time after surgery. At 1 week postoperatively, only the 6 mm IOL with SF6 group had a significantly smaller ACD and IOL position. It is reported that this difference was caused by the buoyancy and surface tension of the gas [12]. In addition, our results suggest that a smaller IOL size is more sensitive to buoyancy. We were concerned that a small ACD (i.e., forward shift of IOL) increases the risk of complications. Posterior synechia or pupillary capture after gas tamponade has been reported [16,17]; thus, complications should be considered in the early postoperative period of phacovitrectomy with SF6, especially in the 6 mm IOL.

At 1 month postoperatively, there was no difference in the ACD, IOL position, or RPE in all the groups. These results were different from those of a previous report [12] that showed that the ACD and IOL position were significantly smaller in the SF6 injection group than those in the cataract surgery-alone group or in the without SF6 injection group at 1 month postoperatively. This difference might be due to the duration of the prone position. In the previous report [12], the prone position lasted 2–7 days, whereas in the present study, the prone position lasted 1–2 weeks. The longer the prone position, the IOL became less affected by the SF6, and we considered that this could have influenced the results. Although several previous reports have reported no difference in IOL-related complications between patients placed in the prone position and those who were not [18,19], differences in the ACD dependent on the prone position should be investigated in the future.

Changes in refractive index due to removal of the vitreous after vitrectomy have been reported [5]. However, in our study, the ACD and IOL position were not significantly different at 1 month postoperatively, regardless of SF6 injection or the size of the IOL. Our study included a wide variety of pathologic types, and the differences could not be obtained because of the large variability in RPE. Some reports indicated that the design and material of the haptics affect the ACD [20,21] and should be investigated for each type of lens in the future.

This study had several limitations. First, the number of cases in each group was relatively small (*n* = 20), which precluded in-depth statistical analysis; hence, future studies with a larger number of cases in each group are necessary. Second, although we examined the lens size, we did not examine the shape and size of the haptics. As differences in the shape and size of the haptic may also affect resistance to gas tamponade, the statistics for each lens might be necessary. Third, patient pathologies were diverse. This might be the reason why no change appeared in RPE at 1 month postoperatively. Future studies with more consistent indicators, such as disease, age, and sex are warranted. Lastly, patients with SF6 tamponade were required to be in the prone position; however, this management is patient-dependent. Therefore, the effect of ACD on the degree of facedown was not considered in this study.

## 5. Conclusions

In the SF6 groups, the ACD and IOL position of the 6 and 7 mm IOL subgroups were significantly smaller than those of the other groups at 1 day postoperatively. At 1 week postoperatively, the ACD and IOL position of the 6 mm IOL with SF6 group was significantly smaller than that of the other groups. Therefore, complications such as synechia or pupillary capture should be considered in the early postoperative period of phacovitrectomy with SF6, especially in the 6 mm IOL. At 1 month postoperatively, there was no significant difference in ACD, IOL position, and RPE between any of the groups.

## Figures and Tables

**Figure 1 jcm-11-06430-f001:**
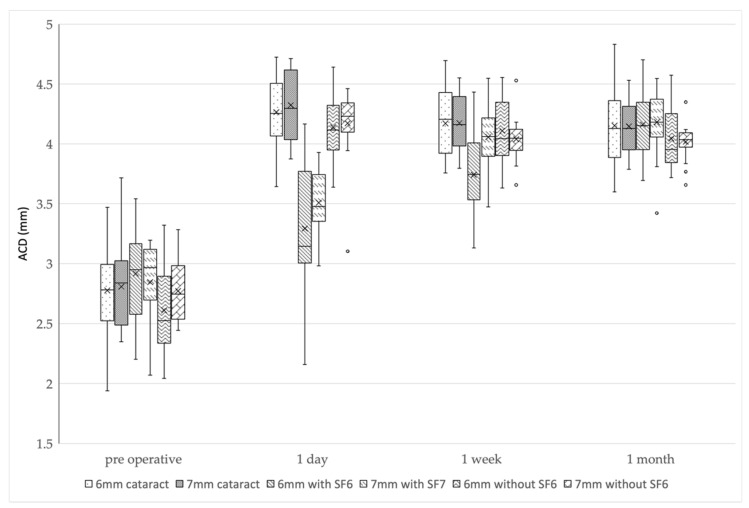
ACD changes in the six groups. The 6 and 7 mm IOL with SF6 groups had significantly smaller ACDs than the other groups at 1 day postoperatively. The 6 mm IOL with SF6 group had significantly smaller ACD than the other groups at 1 week postoperatively. There was no difference in ACD among the groups at 1 month postoperatively. ACD, anterior chamber depth; IOL, intraocular lens; SF6, sulfur hexafluoride; 6 mm cataract, 6 mm IOL cataract surgery-alone group; 7 mm cataract, 7 mm IOL cataract surgery-alone group; 6 mm with SF6, 6 mm IOL with SF6 group); 7 mm with SF6, 7 mm IOL with SF6 group; 6 mm without SF6, 6 mm IOL without SF6 group; 7 mm without SF6, 7 mm IOL without SF6 group.

**Figure 2 jcm-11-06430-f002:**
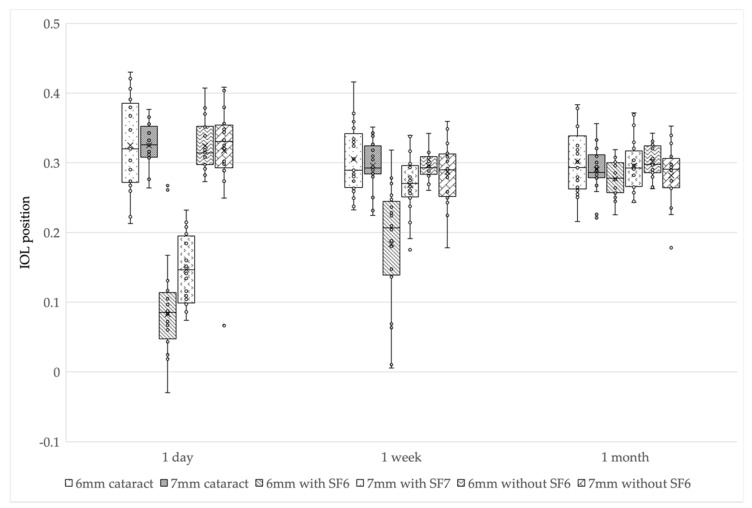
IOL position changes in the six groups. The 6 and 7 mm IOL with SF6 groups had significantly smaller IOL positions than the other groups at 1 day postoperatively. The 6 mm IOL with SF6 group had significantly smaller IOL position than the other groups at 1 week postoperatively. There was no difference in IOL position at 1 month postoperatively. IOL, intraocular lens; SF6, sulfur hexafluoride; 6 mm cataract, 6 mm IOL cataract surgery—alone group; 7 mm cataract, 7 mm IOL cataract surgery—alone group; 6 mm with SF6, 6 mm IOL with SF6 group); 7 mm with SF6, 7 mm IOL with SF6 group; 6 mm without SF6, 6 mm IOL without SF6 group; 7 mm without SF6, 7 mm IOL without SF6 group.

**Table 1 jcm-11-06430-t001:** IOL features.

	XY-1^®^	Clareon^®^	YP2.2R^®^	Tecnis Eyhans^®^ Optiblue^®^	X-70^®^
Shape	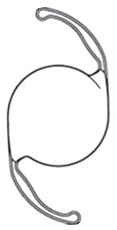	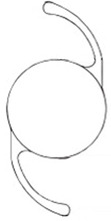	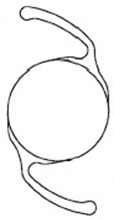	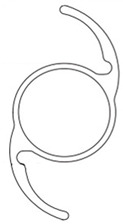	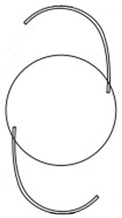
Cross section	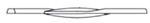	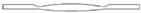	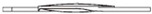	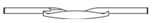	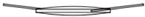
Design	One piece	One piece	One piece	One piece	Three pieces
Overall diameter (mm)	13.0	13.0	13.0	13.0	13.2
Optical diameter (mm)	6.0	6.0	6.0	6.0	7.0

Data were prepared according to the document supplied along with each IOL, including shapes and cross-section figures. IOL, intraocular lens.

**Table 2 jcm-11-06430-t002:** Patient characteristics.

	6 mm Cataract	7 mm Cataract	6 mm with SF6	7 mm with SF6	6 mm without SF6	7 mm without SF6	*p*-Value
**No. of eyes/no. of patients**	20/20	20/20	20/20	20/20	20/20	20/20	
**Age (years)**	73.0 ± 9.6	63.8 ± 7.6	66.0 ± 10.4	65.2 ± 9.4	72.8 ± 6.8	62.4 ± 10.9	<0.01
**Sex, women**	11 (55)	2 (10)	11 (55)	8 (40)	7 (35)	10 (50)	<0.01
**Eyes, right eye**	14 (70)	13 (65)	14 (70)	5 (25)	12 (60)	11 (55)	0.014
**Axial length (mm)**	23.85 ± 1.15	24.0 ± 1.08	25.05 ± 2.4	24.83 ± 1.73	23.84 ± 1.05	23.77 ± 1.20	0.24
**Lens thickness (mm)**	4.58 ± 0.41	4.57 ± 0.54	4.50 ± 0.37	4.52 ± 0.26	4.75 ± 0.43	4.41 ± 0.37	0.19
**BCVA**	0.54 ± 0.59	0.36 ± 0.23	0.41 ± 0.46	0.61 ± 0.50	0.37 ± 0.23	0.92 ± 0.81	0.09

Values are presented as n/n, mean ± standard deviation, or *n* (%). SF6, sulfur hexafluoride; 6 mm cataract, 6 mm IOL cataract surgery-alone group; 7 mm cataract, 7 mm IOL cataract surgery-alone group; 6 mm with SF6, 6 mm IOL with SF6 group; 7 mm with SF6, 7 mm IOL with SF6 group; 6 mm without SF6, 6 mm IOL without SF6 group; 7 mm without SF6, 7 mm IOL without SF6 group; BCVA, best corrected visual acuity.

**Table 3 jcm-11-06430-t003:** ACD outcomes.

	Preoperative	1 Day	1 Week	1 Month	*p*-Value after Operation
**6 mm cataract**	2.78 ± 0.39	4.27 ± 0.27	4.17 ± 0.28	4.15 ± 0.30	0.011
**7 mm cataract**	2.80 ± 0.28	4.33 ± 0.28	4.17 ± 0.20	4.14 ± 0.21	<0.01
**6 mm with SF6**	2.92 ± 0.40	3.29 ± 0.49	3.74 ± 0.36	4.16 ± 0.31	<0.01
**7 mm with SF6**	2.85 ± 0.33	3.51 ± 0.26	4.06 ± 0.25	4.18 ± 0.27	<0.01
**6 mm without SF6**	2.61 ± 0.36	4.14 ± 0.27	4.10 ± 0.29	4.04 ± 0.26	0.038
**7 mm without SF6**	2.77 ± 0.24	4.17 ± 0.28	4.04 ± 0.17	4.02 ± 0.14	<0.01
***p*-Value**	0.61	<0.01	<0.01	0.26	

ACD, anterior chamber depth; IOL, intraocular lens; SF6, sulfur hexafluoride; 6 mm cataract, 6 mm IOL cataract surgery-alone group; 7 mm cataract, 7 mm IOL cataract surgery-alone group; 6 mm with SF6, 6 mm IOL with SF6 group; 7 mm with SF6, 7 mm IOL with SF6 group; 6 mm without SF6, 6 mm IOL without SF6 group; 7 mm without SF6, 7 mm IOL without SF6 group.

**Table 4 jcm-11-06430-t004:** IOL position outcomes.

	1 Day	1 Week	1 Month	*p*-Value
**6 mm** **cataract**	0.33 ± 0.06	0.31 ± 0.05	0.30 ± 0.05	0.012
**7 mm** **cataract**	0.33 ± 0.03	0.30 ± 0.03	0.29 ± 0.03	<0.01
**6 mm** **with SF6**	0.08 ± 0.09	0.18 ± 0.09	0.28 ± 0.02	<0.01
**7 mm** **with SF6**	0.14 ± 0.05	0.27 ± 0.04	0.30 ± 0.04	<0.01
**6 mm** **without SF6**	0.32 ± 0.04	0.29 ± 0.02	0.30 ± 0.02	0.038
**7 mm without SF6**	0.32 ± 0.07	0.29 ± 0.05	0.28 ± 0.04	<0.01
***p*-Value**	<0.01	<0.01	0.45	

IOL, intraocular lens; SF6, sulfur hexafluoride; 6 mm cataract, 6 mm IOL cataract surgery-alone group); 7 mm cataract, 7 mm IOL cataract surgery-alone group); 6 mm with SF6, 6 mm IOL with SF6 group); 7 mm with SF6, 7 mm IOL with SF6 group; 6 mm without SF6, 6 mm IOL without SF6 group; 7 mm without SF6, 7 mm IOL without SF6 group.

## Data Availability

All the data generated or analyzed during this study are included in this published article.

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
