# Peer review of "Temporal Change in Anterior Chamber Depth after Combined Vitrectomy and Cataract Surgery Using Different Sizes of Intraocular Lens"

_jcm, 2022, doi:10.3390/jcm11216430_

Round 1
Reviewer 1 Report
The authors aimed to investigate the temporal changes in the ACD and refractive error of IOL after cataract or phacovitrectomy and they found that there were no difference ACD or IOL position and RPE in all groups. ACD and IOL position gradually deepened with less gas.
The topic of the research is of interest to the refractive surgery community. In this case, the results was that no differences were detected but it is also of interest to research community.
The segmentation into too many groups in one of the main limitation due the 120 eyes divided in 6 six groups
In the introduction, additional and irrelevant information were provided due the potential readers that could be expertise of retina and cataract surgery.
The instrument, fabricant city and country should be included on the methods section
A description of the different between the lenses should be included
Inclusion and exclusion criteria were missing
The information about the IOL position could be described in the detail about the reason of the change, in discussion
On the results the presence of box and plot graphs could improve the visual appearance of the results
The third and fourth paragraph of the discussion is a little bit worldly, please rewrite them to improve the comprehension.
Include new and updated references list
Try to avoid references prior to 2010 when possible
Author Response
Comment #1. The segmentation into too many groups in one of the main limitation due the 120 eyes divided in 6 six groups
Response: We completely agree with this comment. We have added the following statement to the Discussion section.
First, the number of cases in each group was relatively small (n=20), which precluded in-depth statistical analysis; hence, future studies with a larger number of cases in each group are necessary.
Comment #2. In the introduction, additional and irrelevant information were provided due the potential readers that could be expertise of retina and cataract surgery.
Response: Thank you for pointing this out. We have changed and simplified the introduction.
Comment #3. The instrument, fabricant city and country should be included on the methods section
Response: Thank you for pointing this out. We have added the manufacturer’s location (city and country).
Comment #4. A description of the different between the lenses should be included
Response: We have added the IOL features in Table 1.
Comment #5. Inclusion and exclusion criteria were missing
Response: We appreciate this important point. We added the inclusion and exclusion criteria in Section 2.1.
Comment #6. The information about the IOL position could be described in the detail about the reason of the change, in discussion
Response: We appreciate this important point raised by the reviewer. We stated that the ACD and IOL position had the same trend. Hence, we added the reason for the IOL position changes in Section 4 (discussion), paragraph 2.
Comment #7. On the results the presence of box and plot graphs could improve the visual appearance of the results 
Response: We completely agree with this comment. We have revised Figures 1 (ACD changes) and 2 (IOL position) as box and plot graphs.
Comment #8: The third and fourth paragraph of the discussion is a little bit worldly, please rewrite them to improve the comprehension.
Response: Thank you for pointing this out. We have revised the third and fourth paragraphs of Section 4 (Discussion).
Comment #9: Include new and updated references list.
Comment #10: Try to avoid references prior to 2010 when possible
Response: Thank you for these suggestions. We have updated the references list with articles published after 2010.

Reviewer 2 Report
The clinical study investigates changes in anterior chamber depth and refractive preditiction error post cataract surgery with or without the sulfur hexafluoride. The study efficiently, reports significantly smaller ACD and IOL in patients with SF6 injection at day 1 and week 1 and no difference at 1 month between the different group. I recommend the paper to be accepted with minor corrections.
- The overall introduction needs to be more explanatory, highlighting the specifics of the procedure.
- Remove “the” in line 2 of the 2.4 surgical procedure
- Please correct the font differences (mm) in Table 1 headings
- What software was used for statistical analysis?
- Are there any prior studies that highlight the role of prone position that could affect IOL exposure to gas?
Author Response
Comment #1: The overall introduction needs to be more explanatory, highlighting the specifics of the procedure.
Response: Thank you for this suggestion. We have revised and improved the introduction.
Comment #2: Remove “the” in line 2 of the 2.4 surgical procedure
Response: Thank you for pointing this out. We removed the definite article “the.”
Comment #3: Please correct the font differences (mm) in Table 1 headings
Response: Thank you for pointing this out. We have corrected the font in Table 1 headings.
Comment #4: What software was used for statistical analysis?
Response: We used Python’s scipy.stats module. This software is available free of charge and can be used through Python programming.
Comment #5: Are there any prior studies that highlight the role of prone position that could affect IOL exposure to gas?
Response: We could not find any article on the effect of prone position on IOL. Some papers reported no difference in IOL-related complications between the prone and supine postoperative positions in a subanalysis [1,2]. Otsuka et al. [2] investigated ACDs in the prone and supine positions, but they did not mention any difference between the two. We have cited these references to paragraph 4 of Section 4 (Discussion) and added them to the reference list as references 18 and 19, respectively.
In the previous report [12], the prone position lasted 2–7 days, whereas in the present study, the prone position lasted 1–2 weeks. The longer the prone position, the IOL became less affected by the SF6, and we considered that this could have influenced the results. Although several previous reports have reported no difference in IOL-related complications between patients placed in the prone position and those who were not [18,19], differences in the ACD dependent on the prone position should be investigated in the future.
- Shiraki, N.; Sakimoto, S.; Sakaguchi, H.; Nishida, K.; Nishida, K.; Kamei, M. Vitrectomy without prone positioning for rhegmatogenous retinal detachments in eyes with inferior retinal breaks. PLoS ONE. 2018, 13,
- Otsuka, K.; Imai, H.; Miki, A.; Nakamura, M. Impact of postoperative positioning on the outcome of pars plana vitrectomy with gas tamponade for primary rhegmatogenous retinal detachment: comparison between supine and prone positioning. Acta Ophthalmol. 2018, 96, e189–e194.

Round 2
Reviewer 1 Report
The authors solved all the comments proposed in the previous revision
Author Response
Dear Editor and Reviewers:
Thank you for reviewing our manuscript and providing these excellent comments. We have revised the manuscript in accordance with these recommendations. Please find below “point-by-point” responses to the comments and the corresponding changes to the manuscript. These changes are also indicated in red in the “revised manuscript highlighted” file.
Comment #1. What was the clinical implication of these findings?
Response: The main clinical implication in this study is that in 6-mm IOL with SF6 group, the ACD is smaller than other groups up to 1 week postoperatively and which means that complications need to be considered in this period. So, we have added this point in conclusion.
Comment #2. Did the authors notice a significant difference in visual outcomes in 6 and 7 mm IOL groups?
Response: We have additionally examined the point you raised. At one month postoperatively, the BCVA of 6 groups was 0.10±0.25 at 6-mm IOL cataract surgery alone group, 0.21±0.30 at 7-mm IOL cataract surgery alone group, 0.11±0.19 at 6-mm IOL with SF6 group, 0.27±0.41 at 7-mm IOL with SF6 group, 0.27±0.32 at 6-mm IOL without SF6 group, and 0.35±0.29 at 7-mm IOL without SF6 group. And there was significant difference between 6 groups (P=0.03, Kruskal-wallis test). We assume that this difference is due to disease characteristics. For example, PDR patients have poor postoperative vision. Seven PDR patients was included in 7-mm IOL without SF6 group while the 2 PDR patients was included in 6-mm IOL without SF6 group. So, we did not mention BCVA in the text because we have considered it to be not related to refraction.
Comment #3. What was the refractive difference that could gave been attributed to IOL position shift after surgery?
Response: We have additionally examined the point you raised. There was no difference in RPE at 1 month postoperatively in the 6 groups, and we have considered that there was no relationship between IOL position shift and refractive error as shown in the figure below. We have investigated the correlation between IOL position and RPE, and there was no correlation between IOL position and RPE (correlation coefficient = 0.04).
Comment #4. The conclusions are not clearly defined in the abstract or the manuscript.
Response: We appreciate this important point raised by the reviewer. As mentioned in the response to comment #1, we have added the clinical implications to the abstract and conclusion.
